# Siamese Networks for Clinically Relevant Bacteria Classification Based on Raman Spectroscopy

**DOI:** 10.3390/molecules29051061

**Published:** 2024-02-28

**Authors:** Jhonatan Contreras, Sara Mostafapour, Jürgen Popp, Thomas Bocklitz

**Affiliations:** 1Institute of Physical Chemistry (IPC) and Abbe Center of Photonics (ACP), Friedrich Schiller University Jena, Member of the Leibniz Centre for Photonics in Infection Research (LPI), Helmholtzweg 4, 07743 Jena, Germany; jhonatan.contreras@uni-jena.de (J.C.); sara.mostafapourghasrodashti@uni-jena.de (S.M.); juergen.popp@uni-jena.de (J.P.); 2Leibniz Institute of Photonic Technology, Member of Leibniz Health Technologies, Member of the Leibniz, Centre for Photonics in Infection Research (LPI), Albert Einstein Straße 9, 07745 Jena, Germany; 3Institute of Computer Science, Faculty of Mathematics, Physics & Computer Science, University Bayreuth Universitaetsstraße 30, 95447 Bayreuth, Germany

**Keywords:** Siamese networks, machine learning, bacteria classification, Raman spectroscopy

## Abstract

Identifying bacterial strains is essential in microbiology for various practical applications, such as disease diagnosis and quality monitoring of food and water. Classical machine learning algorithms have been utilized to identify bacteria based on their Raman spectra. However, convolutional neural networks (CNNs) offer higher classification accuracy, but they require extensive training sets and retraining of previous untrained class targets can be costly and time-consuming. Siamese networks have emerged as a promising solution. They are composed of two CNNs with the same structure and a final network that acts as a distance metric, converting the classification problem into a similarity problem. Classical machine learning approaches, shallow and deep CNNs, and two Siamese network variants were tailored and tested on Raman spectral datasets of bacteria. The methods were evaluated based on mean sensitivity, training time, prediction time, and the number of parameters. In this comparison, Siamese-model2 achieved the highest mean sensitivity of 83.61 ± 4.73 and demonstrated remarkable performance in handling unbalanced and limited data scenarios, achieving a prediction accuracy of 73%. Therefore, the choice of model depends on the specific trade-off between accuracy, (prediction/training) time, and resources for the particular application. Classical machine learning models and shallow CNN models may be more suitable if time and computational resources are a concern. Siamese networks are a good choice for small datasets and CNN for extensive data.

## 1. Introduction

Bacteria analysis including bacteria detection, identification, discrimination, and antibiotic resistance is important in food safety control, infectious disease prevention, and environmental monitoring [1,2]. Culture-based methods are the most used methods in bacteria detection and identification [3]. However, these methods are slow, labor-intensive, and do not meet requirements for the rapid detection of bacteria. Improved analytical methods such as polymerase chain reaction (PCR) [4,5] and immunological assays [6,7] have been developed to reduce overall testing time with high specificity but have limitations for field testing and require expensive reagents. Therefore, alternative methods to improve testing efficiency are needed.

Raman spectroscopy is a powerful technique for the rapid, sensitive, and non-destructive detection of bacteria. Raman spectroscopy is a powerful technique that uses scattered light to provide a chemical fingerprint of the vibrational modes within the sample’s composing molecules [8,9]. The technique has been applied in numerous fields where identifying unknown substances is crucial, including food quality control [10,11], pharmaceutical drug characterization [12], forensic investigations [13], and bacterial detection [14,15,16]. However, interpreting untargeted spectral data from Raman spectroscopy can be challenging due to the complexity of the data and the properties of biological samples. To address this, chemometric techniques are commonly used to analyze Raman data in the fields of chemistry, biology, and biochemistry [17,18].

Within chemometrics, classical machine learning methods have been widely used for data modeling such as unsupervised clustering, classification, and regression tasks. Dimension reduction, often the initial step in data modeling is crucial in extracting valuable features from the data. Principal component analysis (PCA) and partial least squares (PLS) are two dimension reduction techniques utilized in data modeling [19]. Selecting an appropriate number of components is critical in this area as too few components might not adequately capture the underlying trends in the data. Conversely, an excessive number of components might introduce noise, reducing the effectiveness of the model [20,21]. Following dimension reduction, the processed data can be used as input for various classification models, including linear discriminant analysis (LDA), support vector machine (SVM), and random forest (RF).

The LDA method is a widely utilized classification method that aims to maximize the ratio of between-class variance to within-class variance, thereby enhancing class separability. However, it encounters challenges with small sample sizes, where the number of samples is less than the number of variables and is limited to modeling only linearly separable data. The small sample size issue can be mitigated by applying PCA before LDA, a technique known as PCA-LDA. To address non-linearity, where classes are not linearly separable, kernel functions can be employed to extend LDA’s applicability [22,23].

SVM [24] uses kernel functions to handle nonlinear classification problems by maximizing the interval between samples and decision boundary. However, this method is sensitive to missing data, which can affect its accuracy. Random forests (RFs) [25] are particularly adept at processing large datasets due to their ensemble approach, which combines multiple decision trees to improve classification accuracy and handle noise. However, the algorithm’s efficiency can be compromised by the increased training time associated with a large number of decision trees, and it is possible that the model may overfit when dealing with highly noisy data [26].

A vital step in these classical machine learning methods is spectral pre-processing, which varies depending on the spectrometer and its configuration, which could produce different noise characteristics and artifacts. At the same time, the pre-processing of the signal has the potential risk of introducing further errors and variability [27,28,29,30]. Therefore, deep learning techniques in vibrational spectroscopy have been investigated in recent years. Deep neural networks have many advantages compared to classical machine learning, such as handling large datasets, both pre-process and without pre-processing and achieving superior performance [30,31,32]. Application of different deep learning methods like artificial neural network (ANN), convolutional neural network (CNN), auto-encoder, generative adversarial network (GAN), recurrent neural network (RNN), etc. in Raman spectroscopy data analysis can be found in previous studies [33,34].

The predecessor of deep learning is artificial neural networks (ANNs); however, among deep learning techniques, convolutional neural networks (CNNs) have been commonly used for spectral matching since they can extract fingerprint characteristics effectively, which leads to higher classification accuracy [33]. For example, a CNN is utilized to rapidly identify *Salmonella serovars* [35], distinguish between live and dead *Salmonella* [36], discrimination of clinically significant pathogens [37], discriminate between Carbapenem-resistant and Carbapenem-sensitive *Klebsiella pneumoniae* strains [38], and identify antibiotic resistance and virulence encoding factors in *Klebsiella pneumoniae* [39].

CNNs are trained in a supervised manner and optimized directly for each reference substance or class in the training database, and their performance strongly depends on the amount of training data. Additionally, retraining is needed when the reference database or the training set are modified, which induces impractical computational costs., i.e., the addition of a new class or the availability of more training data [34,40,41]. To address these limitations, a Siamese network has been proposed, which converts the classification problem into a similarity problem and solves the issue of limited available data for training CNNs [42,43,44,45,46,47], where, in many practical cases, only a few spectra are available per substance or class.

Siamese networks aim to determine similarity or dissimilarity between pairs of data points. Unlike CNN classification methods that rely on large datasets for training, Siamese networks work differently. In the training process the model is presented with pairs of examples with indications of whether they belong to the same class (similar) or different class (dissimilar), and the model learns to generate a similarity score or distance measure for each input pairs, providing a better understanding of how closely or distantly related they are. This approach differs from classification strategies that focus on assigning singular labels to individual instances. Siamese networks are more flexible and efficient when data is limited.

Our Siamese network architecture consists of two identical convolutional neural networks (CNNs) that share the same weights. The extracted features are fed into a final dense layer that calculates the similarity metric between the two spectra. This function can determine how similar or dissimilar two spectra are. In order to introduce CNNs, the model description section aims to provide a comparison between classical machine learning methods and deep learning techniques in Raman spectroscopy analysis. The comparison includes the accuracy metrics and the computational costs of these methods. The paper will also discuss potential future directions in Raman spectroscopy, mainly using Siamese networks over conventional CNNs in tasks where data availability is limited.

## 2. Results and Discussion

In this study, different classical machine learning and deep learning methods for analysis of Raman spectra from bacteria datasets are investigated. A workflow of this study is shown in Figure 1.

### 2.1. Classification on a Six-Bacterial-Species Dataset

This Raman spectral dataset contains 5420 single bacteria spectra, which include six bacterial species and around 900 spectra per class showing a balanced dataset. The samples were cultivated in nine independent biological replicates. We use two batches of cross-validation to evaluate the stability of the results. In every cross-validation fold, the test set contains two batches, and the rest is used as a training and validation set (70% training and 30% validation). Table 1 presents the performance metrics of five different classical machine learning models: PCA-LDA, PCA-SVM, PLS-DA, and PCA-RF as classical machine learning models, and Shallow CNN, Deeper CNN, Siamese-model1, and Siamese-model2 as deep learning models. The models are evaluated based on sensitivity, specificity, training time, and the number of parameters, where sensitivity is the ability of the model to identify positive instances correctly and specificity indicates the false positive rate of the model. The number of parameters in PCA-LDA, PCA-SVM, and PCA-RF shows the number of principal components for dimensional reduction and in PLS-DA is the number of latent variables in PLS decomposition.

The fitting time for the classical machine learning models cannot be compared directly with the CNN models. Although we set the maximum number of epochs for all networks to 200 with early stopping based on validation data and patience of 20 epochs, the actual number of epochs required for convergence varies depending on several factors. Random initialization, learning rate, batch normalization, and model complexity are among the factors that can influence the number of epochs needed for convergence. Therefore, we report the duration of an entire training model (200 epochs), it is essential to consider that the actual number of epochs required may vary depending on the specific model and its parameters.

The prediction time reported in Table 1 corresponds to the time needed to predict a single bacteria spectrum. Classical machine learning model prediction is straightforward and recommended when time is the most critical aspect as the prediction of a single spectrum took only around 0.0002 s. Table 1 displays high specificity values for all models, indicating correct identification of negative results. However, sensitivity is better with any of the CNN methods as the mean sensitivity of classical machine learning methods is approximately 80%.

Prediction times for the shallow and deeper CNN models are constant. However, for the Siamese networks, during testing, we averaged N times over k-shots. This means that for each test spectrum, k samples per class are randomly selected, this is repeated N times, and the average is calculated. Table 1 presents the prediction time for two values of k-shots for N = 1. Predicting 50 shots takes a significant amount of time as it involves comparing our input spectrum 300 times with 50 samples for each of the six classes, which becomes even more problematic when the number of classes is higher. However, we have found experimentally that the values of k can range between 10 and 30, with N equal to 1. During our experiments, we observed that excluding the distance predictions (output of the Siamese networks) falling below the 10th percentile and above the 90th percentile instead of a simple mean value yields a more accurate prediction.

The number of parameters indicates the complexity of the models and should be compared as well. Classical machine learning methods have only 20 principal components, thus only 21 parameters are fitted, with 20 weights and the classification threshold, whereas all other models have millions of parameters, except for the shallow CNN, which has only 14.7 K parameters. The two highest sensitivity values are for the deeper CNN and Siamese-model2. Although the CNN sensitivity is 0.52% higher than that of the Siamese network, the standard deviation provides insight into the more general performance of the model across the 36 different cross-validated models, indicating that Siamese-model2 has a more stable behavior. In summary, the CNN-based approaches perform similarly, with 82.80 ± 13.54 (shallow CNN), 84.13 ± 12.30 (deeper CNN), 82.65 ± 4.39 (Siamese-model1), and 83.61 ± 4.73 (Siamese-model2) mean sensitivity.

### 2.2. Pre-Training and Fine-Tuning on 15 Bacterial Strains

In this section, we utilized 50% of the classes from the public Raman spectra dataset [14] to improve the visualization embeddings created by the Siamese network. Next, we tested the effectiveness of our methodology by analyzing all 30 available in a balance and imbalance scenario.

Siamese-model2 was employed due to its superior performance as determined by our evaluation metrics and considerations, and to evaluate the effectiveness of our approach. We pre-trained the Siamese network using only 50% of the available classes of the reference dataset, which allowed us to emulate a more realistic scenario.

Training first the sub-network of the Siamese network separately speeds up the process, but it is also possible to train the network end-to-end. This subnetwork was trained using the triplet semi-hard loss function, which utilizes triplets, where the negative example is farther from the anchor than the positive one, to create an embedding vector separating the classes. To identify these triplets efficiently, we used Semi-Hard online learning in each batch, allowing us to optimize the subnetwork’s performance.

Appendix A shows the learned embedding projections of three PCA components that capture only 52% of the variance in the testing data, comprising 15 classes. Each point in the plot is color-coded based on the ground truth label, and the corresponding number is also shown. We further highlight some of the clusters that are well separated from the other classes in Appendix A. While it is possible to differentiate at least half of the classes based on the embeddings, the variance between the other classes is low, and the intra-class variance is high. Thus, even for similar Raman data trained on a comparable spectrum, fine-tuning is necessary to achieve optimal performance.

After fine-tuning the subnetwork using half of the classes of the fine-tuning dataset, we obtained a second embedding vector. Appendix A displays the embedding projections of three PCA components that capture 47% of the variance in the testing data. Each point in the plot is color-coded based on the ground truth label, which may result in some colors being repeated. We observe that the majority of the classes can be distinguished from each other based on the embedding, indicating that the fine-tuning process has enhanced the subnetwork. Figure 2 displays the confusion matrix derived from the testing dataset, where the model was tested with ten shots. Analysis of the confusion matrix reveals an average sensitivity of 72.0%, an average precision of 76.1%, and an F1 score of 0.71. The F1 score integrates precision and sensitivity into a single metric to gain a better understanding of model performance. The F1 score is calculated by:$$F1\;score=\frac{2\times precision\times sensitivity}{precision+sensitivity}$$

### 2.3. Classification on a 30-Bacterial-Strain Dataset

In this section, Siamese model2 is pre-trained using the reference data dataset [14], and subsequently we examine two distinct fine-tuning scenarios. The initial scenario entails pre-training on 30 classes, followed by training the Siamese network with a balanced dataset, consisting of 100 spectra for each class. In contrast, the second scenario adopts an unbalanced dataset approach, where half of the classes are represented by 100 spectra each, and the remaining half by only 30 spectra each. Moreover, the pre-training process incorporates only 50% of the classes, a strategy specifically designed to assess the model’s proficiency in managing scenarios characterized by limited data availability.

Figure 3 showcases the confusion matrices for the two scenarios under investigation. This detailed representation highlights the classification outcomes when training the model under balanced versus unbalanced conditions. For the testing phase, ten shots were employed.

In the balanced scenario, utilizing the full dataset yielded a notable average sensitivity of 80.0%, an average precision of 82.0%, and an F1 score of 80.0% For the unbalanced scenario, the model achieved an average sensitivity of 73.0%, an average precision of 78.1%, and an F1 score of 72.9%. These results are particularly significant considering the challenging conditions: the absence of certain classes during pre-training and the limited sample size in the fine-tuning stage. Despite these constraints, the Siamese model attained a commendable prediction accuracy of 73.0%. The Siamese model shows potential in complex scenarios, indicating an ability to address new data classes and manage limited data, which suggests its adaptability and robustness for practical applications.

### 2.4. Rank-2 Accuracy

Siamese network prediction conducts a comparative analysis for each sample against ten distinct reference bacteria from each of the 30 classes. Figure 4 illustrates this comparison through two misclassified spectra. It presents a grouped bar chart showcasing the weighted distance metric between the input spectra; specifically, (a) *S. lugdunensis* and (b) *E. coli*2, in relation to all reference bacteria. In this chart, a greater distance signifies reduced similarity between the spectra, whereas a smaller or negative distance denotes higher similarity. Figure 4 uses green to denote the true class and red to indicate the incorrectly predicted class. The model focuses exclusively on these two specific classes in the scenarios presented, effectively disregarding the other 28. It is important to note that this model’s primary function is not to classify data in the conventional sense but rather to compare input data against a reference spectrum, as reflected in its selective identification process. Furthermore, Appendix A graphically presents the distribution of the weighted distance metric, contrasting correct versus incorrect predictions. This visualization clearly demonstrates that samples with incorrect predictions tend to cluster at higher distances, while those correctly classified generally align with lower distance values.

In the previous section, we described a model trained on a balanced dataset that achieved a notable classification accuracy of 80%. However, given the complex and diverse nature of bacteria strains, we incorporated a refined metric called “Rank-2 accuracy”. It considers the accuracy of the second-highest prediction along with the top choice. This metric is relevant when using a Siamese network, which solves a similarity problem, especially in scenarios with 30 different bacterial classes, because it better evaluates the model’s performance. As summarized in Table 2 which also includes Rank-3 accuracy, Rank-1 accuracy stands at 80.26% with 2408 correct classifications, Rank-2 accuracy enhances the model’s precision to 90.26% for an additional 300 spectra, and Rank-3 accuracy further elevates this metric to 93.46% for 96 spectra.

## 3. Materials and Methods

### 3.1. Data Description

The dataset utilized in this study comprised Raman spectra obtained from six distinct bacterial species: *Escherichia coli* DSM 423, *Klebsiella terrigena* DSM 2687, *Pseudomonas stutzeri* DSM 5190, *Listeria innocua* DSM 20649, *Staphylococcus warneri* DSM 20316, and *Staphylococcus cohnii* DSM 20261. All bacterial species were obtained from Deutsche Sammlung von Mikroorganismen and Zellkulturen GmbH (DSMZ) and were cultivated in nine independent biological replicates.

The Raman spectra were acquired using a Raman microscope (Bio Particle Explorer, rap.ID Particle Systems GmbH). The spectral data were collected over a range of 240 to 3190 cm^−1^. The dataset contains 5420 preprocessed spectra with 584 wavenumbers. The number of spectra per class is around 900 spectra showing a balanced dataset. Figure 5 shows the mean spectra for each class. More details about samples and the Raman spectrometer can be found in ref. [48].

Prior to analysis, the Raman spectra underwent preprocessing steps, cosmic spikes removal, wavenumber calibration, spectra alignment, and baseline correction. The preprocessed spectra were then used as inputs for the classification algorithms. The efficacy of various classification algorithms was evaluated utilizing cross-validated weighted accuracy.

We analyzed a second publicly available Raman spectra dataset [14], including 30 common bacterial pathogens. Appendix A describes the species name, label, and isolate code. These pathogens were chosen as they are responsible for most infections observed in intensive care units worldwide. The researchers deposited bacterial cells onto gold-coated silica substrates to create the reference database. They collected spectra from monolayer regions of each strain, ensuring high-quality Raman spectra with minimal interference.

The dataset comprises three subsets: a reference dataset, a reference-finetune dataset, and a test dataset. The reference dataset contains 60,000 evenly distributed Raman spectra from 30 bacterial strains, with 2000 spectra per class. These spectra were used to pre-train machine learning models [14,45]. The reference-finetune dataset was employed to fine-tune the models, while the test dataset was used to evaluate the performance of the models. The reference-finetune dataset and a test dataset contain 3000 spectra representing all 30 bacterial strains with 100 spectra per class. Figure 6 shows the mean spectra for each of the 30 bacterial strains. The spectral data were collected over 381.98 and 1792.4 cm^−1^, distributed uniformly over 1000 wavenumbers. As preprocessing, the spectra intensity was individually normalized between 0 and 1.

### 3.2. One-Dimensional Convolutional Neural Network Model Description

One-dimensional convolutional neural networks (1D-CNNs) have become a popular tool for processing and analysis of sequential data in various domains, such as speech recognition [49], music analysis [50], and financial forecasting [51], for their ability to extract relevant features from sequential data. In a 1D-CNN, the input data is represented as a sequence of values.

Figure 7 illustrates the network architecture for the two 1D-CNN models that are used in this study. In the case of Raman data, the input contains the entire spectrum, where each point represents an intensity in a specific wavenumber. The network then applies a set of filters (kernels) to the input data, with each filter moving across a small window of adjacent data points at a time. The filters perform convolutions, which extract important features or patterns. These features are then passed through additional convolutional layers to create increasingly complex representations of the original input sequence. Additionally, a non-linear activation function (Leaky ReLU) introduces nonlinearity to the model after each layer. Finally, the output of the last convolutional layer is fed through a set of fully connected layers, which perform the final classification followed by SoftMax activation that normalizes the k-dimensional output vector of real values with different dynamic ranges to real values in the range [0,1], that sum to 1 and can be viewed as a probability output.

Figure 7 presents the general concept of a 1D CNN. However, the selection of the optimal values of the hyperparameters, such as the number of layers, kernels, kernel size, dropout value, activation function, batch normalization decay, and momentum, depends on the dataset. Typically, those values are selected through trial and error. We report the results for two configurations, a shallow CNN model (Figure 5 above) composed of a convolutional layer and a dense layer, and a deeper CNN formed by three convolutional layers and three dense layers (Figure 5 below). A more detailed review of these CNN architectures which were built based on Tensorflow framework in Python is shown in Table 3. The shallow CNN model includes a convolutional, a pooling, and a dense layer. The Raman spectrum as input imported into the 1D-CNN model. The convolutional kernel with size 10 × 1 and step size 5 are used for feature extraction. This procedure is followed by a batch normalization (BN) layer. The Leaky ReLU function is used as the activation function, which can help the network learn complex data, improve nonlinear modeling capabilities of the network, and provide more accurate predictions. The pooling layer reduces the dimensionality of the feature vector, enhances the robustness of the network, and obtains lower resolution feature data. The data from pooling layer are input to a fully connected (Dense) layer for classification. The difference between shallow CNN and deeper CNN models is in the number of convolutional, pooling, dense layers, and kernel size.

### 3.3. One-Dimensional Siamese Network Model Description

The Siamese network is a type of deep learning architecture specifically designed for solving problems related to similarity and distance. Figure 8 shows that the architecture of this network consists of two identical sub-networks with the same weights and architecture. Each sub-network takes in a Raman spectrum as input and produces feature vector embedding that captures the essential information in a higher dimensional space. The two feature vectors *f*_1_ and *f*_2_ are then compared using a learnable weighted distance metric:$$d=w^{T}\cdot \|f_{1}-f_{2}\|$$
where the weights of the metric are learned from data during the training process. In our network, it is implemented using a fully connected layer followed by a sigmoid activation. In other words, the metric is not fixed but can adapt to the specific characteristics of the data being analyzed. It measures the distance or similarity between two spectra, the output of the Siamese network is a binary classification, which reveals whether the two input spectra belong to the same class or not.

### 3.4. One-Dimensional Siamese Network Model Training

Since the dataset size is small, we decided not to create a paired spectra dataset and instead generated them randomly during training, which allowed us to reduce overfitting. Given spectrum $x_{i}$, another reference spectrum $x_{j}\;$is randomly selected from the training data set with its corresponding class label. The paired label $y_{i}$ is generated as follows: if the input spectra belong to the same category, $y_{i}$ is set to zero, indicating poor distance and similarity. On the other hand, if the input spectra belong to different classes, $y_{i}$ is set to one, indicating significant distance and negative similarity. In this way, the problem is transformed into a binary classification in which we use the binary cross entropy as the loss function.
$$L=-\frac{1}{N}\sum_{i=1}^{N}y_{i}\log p_{i}+\left(1-y_{i}\right)\log\left(1-p_{i}\right)$$
where N is the number of paired spectra, $p_{i}$ is the outcome of the Siamese network and $y_{i}$ is the paired label. The models were trained with the Adam optimizer with a learning rate of $6\times 10^{-5}$ and batch size of 64. We report the results for two configurations. For both cases, the feature extraction part is identical. The difference lies in the learnable distance function. In the first case, we use only one dense layer, and in the second configuration, we use a more complex function that uses three dense layers, as shown in Table 3.

## 4. Conclusions

As discussed earlier, to utilize Raman spectroscopy in real world applications like diagnostics, chemometrics and machine learning are needed. In this contribution different classical machine learning concepts are compared, which feature different properties. Table 1 presents the results of several classical machine learning models trained and evaluated on a six-bacterial-species dataset, including PCA-LDA, PCA-SVM, PLS-DA, PCA-RF, shallow CNN, deeper CNN, and two variants of the Siamese model. The models were compared based on mean sensitivity, training time, prediction time, and number of parameters. Based on the application scenario, the restrictions of the application and the available data different models perform best.

If the goal is to maximize model performance, the best choice would be the Siamese model 2, which achieved the highest mean sensitivity of 83.61 ± 4.73. However, if time and computational resources are a concern, the classical machine learning and shallow CNN models would be more suitable, as they required significantly less training and prediction time and contained fewer parameters than the deeper CNN models. Additionally, if the dataset is small, the shallow CNN model may be a better option, as it achieved reasonable accuracy while requiring less training time and having fewer parameters than the deeper CNN and Siamese models. Overall, the choice of model should depend on the specific trade-off between model performance, training/prediction time, and resources that are most important for a particular application [52].

In addition, Siamese-model2 was utilized to classify 30 bacterial strains. The sub-network was pre-trained separately using the triplet semi-hard loss function to generate an efficient embedding vector to separate the classes. We learned the learnable weighted distance metric using the fine-tuning dataset, which improved the model’s ability to differentiate between bacterial strains converting the classification problem into a similarity problem. To better emulate real-world conditions, only half of the available classes from the reference dataset were utilized to pre-train the model. We fine-tuned Siamese-model2 using a limited and highly unbalanced dataset. This presented a challenging task, as 15 classes had 100 samples each, while the remaining 15 classes only had 30 samples each. Nevertheless, such situations are often encountered in medical diagnostics. Despite these difficulties, Siamese-model2 demonstrated remarkable performance, achieving a prediction accuracy of 73%. Our findings highlight the model’s ability to generalize to new classes and handle limited data scenarios effectively. These properties make Siamese-model2 a versatile and robust tool, with immense practical applications. Furthermore, using all available data, we obtained a significantly higher accuracy of 80.04% for 30 classes, further underscoring the potential of Siamese-model2 in bacterial strain identification. With further improvements and optimizations, this approach could lead to more accurate and efficient bacterial classification, aiding in the field of microbiology and disease diagnosis.

## Figures and Tables

**Figure 1 molecules-29-01061-f001:**
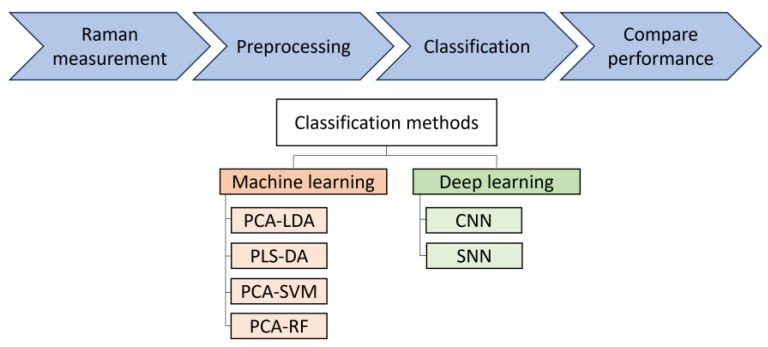
Bacteria Raman data analysis workflow. Different classical machine learning and deep learning method performance is investigated. PCA-LDA: principal component analysis–linear discriminant analysis; PLS-DA: partial least squares–discriminant analysis; PCA-SVM: PCA–support vector machine; RF: random forest; CNN: convolutional neural network; SNN: Siamese neural network.

**Figure 2 molecules-29-01061-f002:**
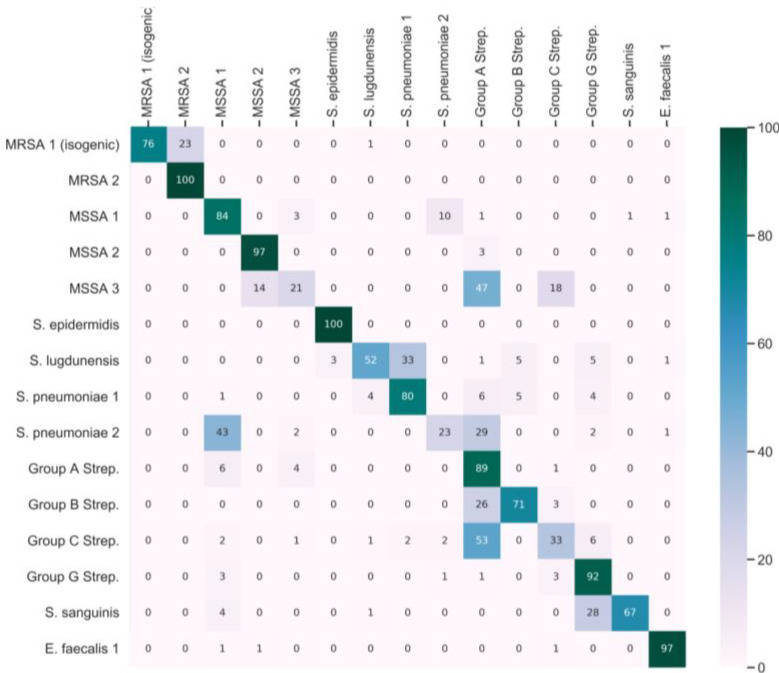
Confusion matrix for the Siamese network. Testing data across the first 15 bacteria strains of a 30-bacterial-strain dataset.

**Figure 3 molecules-29-01061-f003:**
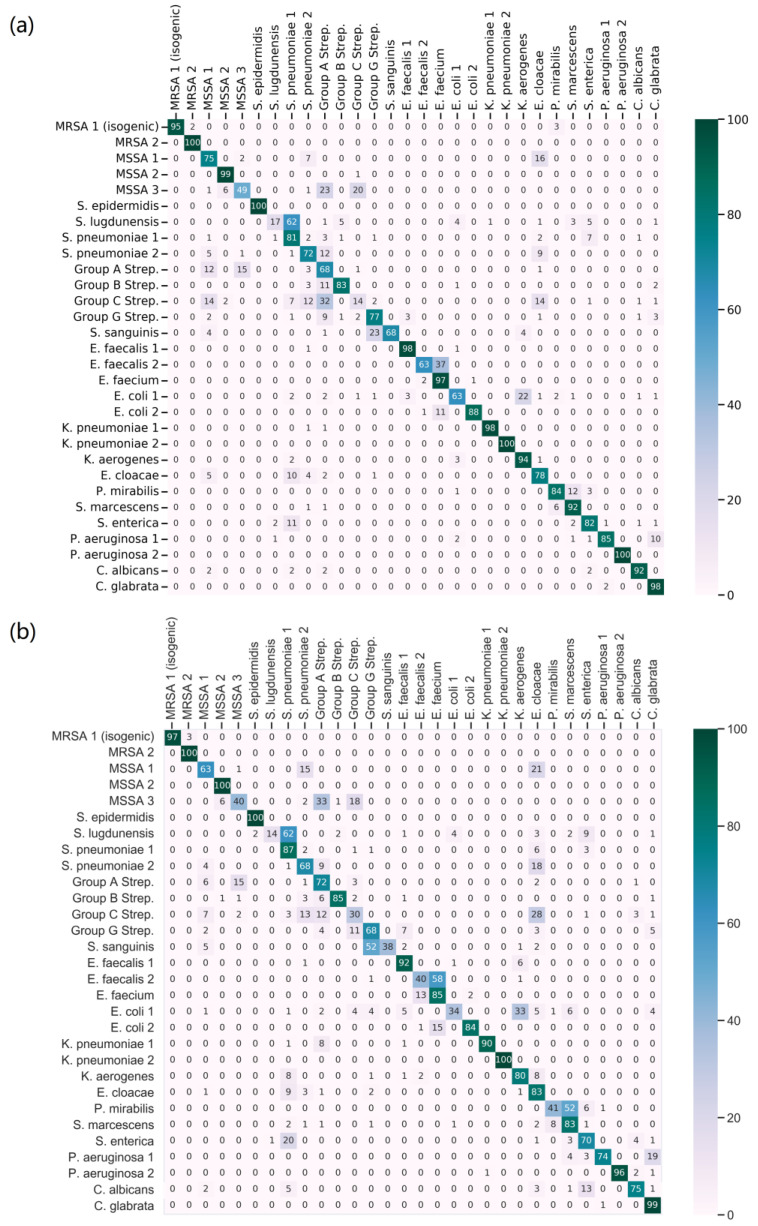
Comparative performance evaluation of Siamese model2: Detailed confusion matrices illustrating classification outcomes in balanced (**a**) vs. unbalanced (**b**) training scenarios.

**Figure 4 molecules-29-01061-f004:**
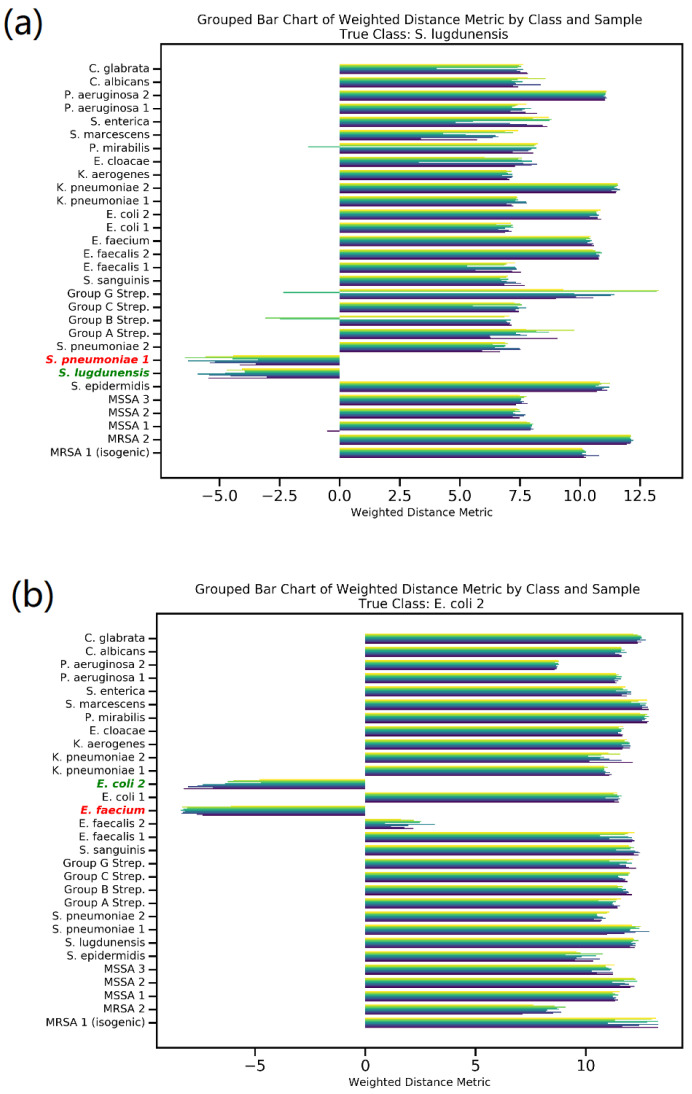
Comparison of misclassified spectra (**a**) *S. lugdunensis* and (**b**) *E. coli* 2. Using weighted distance metrics and grouped bar chart illustration of similarity measures in the 30-class Siamese model2.

**Figure 5 molecules-29-01061-f005:**
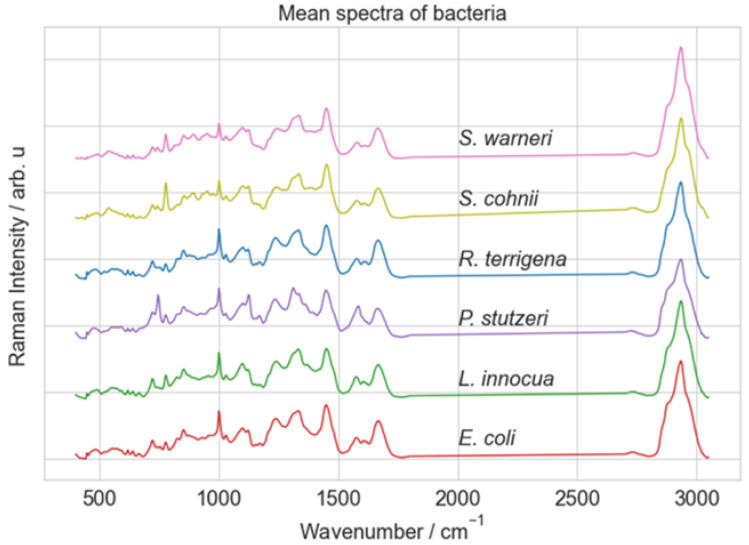
Mean spectra of Raman spectra dataset [48] obtained from six distinct bacterial species.

**Figure 6 molecules-29-01061-f006:**
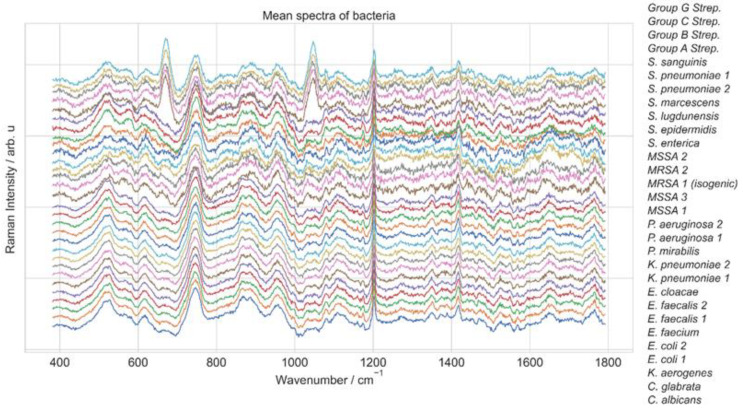
Mean spectra of Raman spectra dataset [14] obtained from 30 distinct bacterial species.

**Figure 7 molecules-29-01061-f007:**
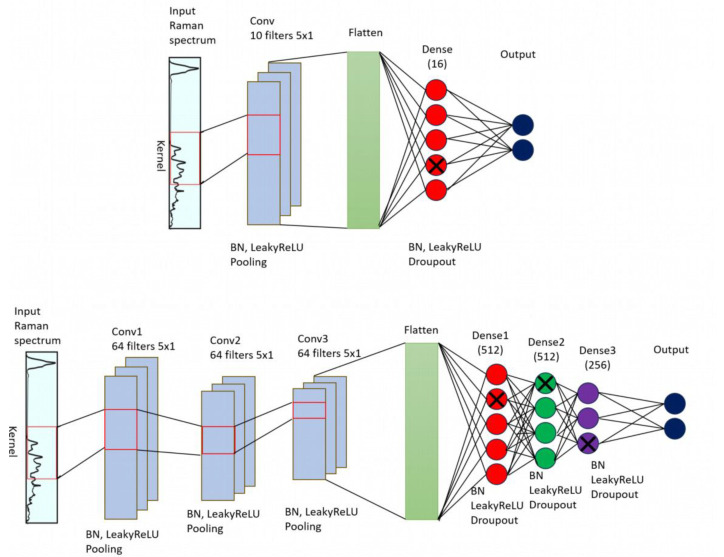
CNN architecture for Raman spectrum classification. It is composed of a set of convolutional layers that extract features, followed by fully connected layers that perform the classification. Above: shallow CNN with one convolutional and one dense layer, below: deeper CNN with three convolutional and three dense layers.

**Figure 8 molecules-29-01061-f008:**
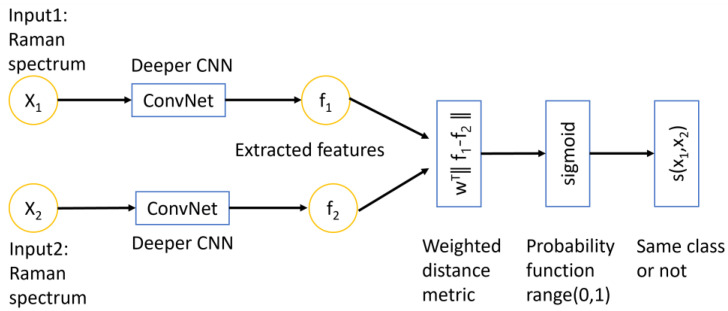
The conceptual architecture of a Siamese network. The inputs are two Raman spectra that analyzed by two identical CNN models. The convolution network is the deeper CNN model that was introduced in previous section. The extracted features are compared using a learnable weighted distance metric and followed by a sigmoid function which predicts the probability between the range of 0 and 1. The output determines whether the two inputs belong to the same class or not (the two inputs are similar or not).

**Table 1 molecules-29-01061-t001:** Mean sensitivity across 36 cross-validated models on a six-bacterial-species dataset, including training time, prediction time for one sample, and the number of parameters.

	Sensitivity (%)	Specificity (%)	Training Time (s)	Prediction Time (s) One Sample		Number Parameters
PCA-LDA	79.85 ± 4.01	95.97 ± 0.80	1.79	0.0002		21
PCA-SVM	80.51 ± 4.75	96.10 ± 0.95	8.65	0.0002		21
PLS-DA	78.58 ± 3.81	95.72 ± 0.76	5.27	0.0002		21
PCA-RF	79.15 ± 4.80	95.82 ± 0.95	98.57	0.0003		21
Shallow CNN	82.80 ± 13.54	96.52 ± 0.89	800	0.040		14.7 K
Deeper CNN	84.13 ± 12.30	96.90 ± 0.83	800	0.047		19.6 M
				k = 10	k = 50	
Siamese model1	82.65 ± 4.39	96.62 ± 0.82	2000	0.070	0.105	19.6 M
Siamese model2	83.61 ± 4.73	96.75 ± 0.92	2000	0.072	0.185	19.6 M

**Table 2 molecules-29-01061-t002:** Hierarchical accuracy metrics, Siamese model2 performance across Ranks 1–3. Shows accuracy percentages and counts of classified instances. Testing data across the 30 bacteria strains.

	Overall Accuracy	Count
Rank-1	80.26	2408
Rank-2	90.26	300
Rank-3	93.46	96

**Table 3 molecules-29-01061-t003:** Comparison of the architectures of the deep learning models used within this article.

	Convolutional Layers	Fully Connected Layers
Shallow CNN	Conv (10,5) BatchNorm + LeakyReLU + pooling	Dense (16) BatchNorm + LeakyReLU + Dropout
Deeper CNN	3 × Conv (64,5) BatchNorm + LeakyReLU + pooling	2 × Dense (512) + Dense (256) BatchNorm + LeakyReLU + Dropout
	Feature vector embedding	Learnable distance metric
Siamese model1	Deeper CNN	Dense (1) Sigmoid
Siamese model2	Deeper CNN	Dense (64) + Dense (16) + Dense (1) Sigmoid

## Data Availability

The data used in this study is public and available in the cited references.

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
