# Peer review of "Siamese Networks for Clinically Relevant Bacteria Classification Based on Raman Spectroscopy"

_molecules, 2024, doi:10.3390/molecules29051061_

Round 1

Reviewer 1 Report

Comments and Suggestions for Authors

This study focuses on the use of Siamese model2 to classify bacterial strains by their Raman spectra. The article compares the Siamese network with conventional machine learning methods and deep learning techniques in Raman spectroscopy analysis.  It discusses the advantages of Siamese networks in tasks with limited data availability and presents experimental results on different datasets.  The results show that the Siamese model achieves acceptable prediction accuracy even under difficult conditions.

The following issues need to be fixed before the paper can be accepted:

1.     Are the models used in this study, such as CNN, Siamese-model2, etc., have been optimized for hyperparameters? If yes, optimization procedures should be included and optimization results should be presented.

2.     The study needs to add some comparisons with other machine learning or neural network models results as a baseline.

3.     Similar to the previous issue, a comparison of results with other published methods is missing from the manuscript.

4.     Fig 1 and Fig 2 are Confusion Matrix for Siamese network for all classes, showing good results. A more detailed analysis of why certain bacterial strains are higher, or lower, in accuracy is expected.

5.     In line 81, " The extracted features are fed into a final dense layer that calculates the similarity metric between the two spectra " is mentioned, but the relationship between bacterial strains similarity and the model's performance in predicting classification is not analyzed in the rest of the study.

6.     In the Conclusion, it is mentioned that different sizes of datasets are suitable for different models. Therefore, the authors are expected to analyze the relationship between the size of the data and the results of the model be supplemented to support this statement.

7.     The authors are expected to provide methods for the visualization of the specific results of classification using the model.

8.    In the beginning of the Results, a figure of workflow can be provided to visualize the logic of this study.

9.  In the introduction, the authors are encouraged to include more examples of CNN applied to the field of chemistry and life sciences. One good example is the study completed by the Rajan group (https://doi.org/10.1021/acs.jcim.0c01409)

Author Response

We want to express our gratitude for providing us with the chance to enhance the quality of our submitted manuscript. The reviewers constructive and insightful comments were greatly appreciated. Please find attached our detailed responses to all queries, addressing each point raised. We have put forth our best efforts to enhance the manuscript and sincerely value your diligent input. We hope that our revisions will meet with your approval. 

Reviewer1:

This study focuses on the use of Siamese model2 to classify bacterial strains by their Raman spectra. The article compares the Siamese network with conventional machine learning methods and deep learning techniques in Raman spectroscopy analysis.  It discusses the advantages of Siamese networks in tasks with limited data availability and presents experimental results on different datasets.  The results show that the Siamese model achieves acceptable prediction accuracy even under difficult conditions.

The following issues need to be fixed before the paper can be accepted:

  1. Are the models used in this study, such as CNN, Siamese-model2, etc., have been optimized for hyperparameters? If yes, optimization procedures should be included and optimization results should be presented.

We appreciate the reviewer's insightful inquiry regarding the optimization of hyperparameters for the models used in our study, such as CNN and Siamese-models. This question underscores the importance of transparency and rigor in the presentation of our machine learning methodology. In response to this, we have mentioned in line 254: “However, the selection of the optimal values of the hyperparameters, such as the number of layers, kernels, kernel size, dropout value, activation function, batch normalization decay, and momentum, depends on the dataset. Typically, those values are selected through trial and error. We report the results for two configurations, a shallow CNN model (Figure 5 above) composed of a convolutional layer and a dense layer, and a deeper CNN formed by three convolutional layers and three dense layers (figure 5 below), as shown in Table 2.”

The process and rationale behind the hyperparameter selection is primarily determined through trial and error. We illustrated two scenarios in the manuscript. In the first scenario, a network is at its smallest viable size and yields satisfactory results. The second scenario involves a significantly larger network that achieves superior outcomes. However, scenarios with either too small or excessively large networks are omitted due to their poor performance. An undersized network struggles to capture the input's variability due to insufficient feature extraction capabilities. On the other hand, an oversized network may encounter the vanishing gradient issue, which compromises its effectiveness. For the "deeper CNN", minor adjustments to some hyperparameters, such as the number of kernels, may marginally enhance performance depending on the training data. However, kernel initialization is inherently random, so improvements are not consistently replicable across multiple iterations. To maintain fairness in our evaluation, we use an identical model configuration across all 36 iterations of the cross-validation process. In parallel, the Siamese network adopts a similar architecture to the deeper CNN. This similarity allows for an unbiased comparison between the two methodologies, independent of any parameter optimization for either model. Finally, the models utilize Sparse Categorical Cross entropy for the loss function. It also uses the Adam optimizer with a learning rate of 1e-5, which optimizes the learnable parameters based on the training data.
This explanation aims to clarify the optimization procedures and substantiate the results presented in our study, ensuring the robustness and reliability of our proposed models.

  1. The study needs to add some comparisons with other machine learning or neural network models results as a baseline.

Thank you for emphasizing the necessity of incorporating comparative analysis with other machine learning or neural network models to establish a baseline in our study. This perspective is indeed valuable, highlighting the importance of contextualizing our findings within the broader spectrum of existing methodologies. In accordance with this suggestion, we have included additional comparisons of other machine learning methods including PLS-DA, PCA-SVM and PCA-RF. The classification parameters of all models are shown in Table 1. We trust that this inclusion will provide a more comprehensive understanding of our model's performance in relation to established methods.

  1. Similar to the previous issue, a comparison of results with other published methods is missing from the manuscript.

We are grateful for the reviewer's insightful observation regarding the comparison of our results with other published methods. We have addressed this point by incorporating results of other machine learning methods including PLS-DA, PCA-SVM and PCA-RF. The classification parameters of all models are shown in Table 1. This comparative approach offers a clearer perspective on the advantages of Siamese Networks for bacteria strains classification.

  1. Fig 1 and Fig 2 are Confusion Matrix for Siamese network for all classes, showing good results. A more detailed analysis of why certain bacterial strains are higher, or lower, in accuracy is expected.

We appreciate the reviewer's request for a more detailed analysis concerning the prediction among different bacterial strains as depicted in Fig 1 and Fig 2. Recognizing the importance of a thorough exploration into the factors influencing these results, we have undertaken a more comprehensive examination. We enhanced sections 2.3 and 2.4 to include a more detailed analysis of the results. We also include in the supplementary material Table S1, detailing species names, figure labels, and isolate codes. Furthermore, Figure S3, in the supplemental material, illustrates the mean spectra and standard deviation from training data for three bacterial strains, highlighting both the similarities and distinct patterns among bacterial strains, notably between two E. coli variants and K. aerogenes, indicating subtle but significant differences crucial for accurate classification and analysis. This enhancement in our analysis aims to provide a clearer understanding of the factors that contribute to the accuracy variations observed in the classification of bacterial strains.

  1. In line 81, " The extracted features are fed into a final dense layer that calculates the similarity metric between the two spectra " is mentioned, but the relationship between bacterial strains similarity and the model's performance in predicting classification is not analyzed in the rest of the study.

Thank you for drawing attention to the relationship between the similarity of bacterial strains and the model's performance in predicting classification, as mentioned in line 81. Your observation rightly points out an area of our study that merits deeper analysis. In response, we have conducted a more detailed examination to bridge this gap.

For the Siamese Networks, during testing for each test spectrum, 10 samples per class are selected, and the average is calculated. During our experiments, we observed that excluding the distance predictions (output of the Siamese networks) falling below the 10th percentile and above the 90th percentile instead of a simple mean value yields a more accurate prediction. We include in the supplementary material Figure S4, which presents the distribution of the weighted distance metric, contrasting correct versus incorrect predictions. This visualization demonstrates that samples with incorrect predictions tend to cluster at higher distances, while those correctly classified generally align with lower distance values. Through this additional analysis, we aim to provide a more comprehensive understanding of how the model leverages similarity metrics between spectra for accurate bacterial strain classification, thereby enhancing the robustness and interpretative depth of our study's findings.

  1. In the Conclusion, it is mentioned that different sizes of datasets are suitable for different models. Therefore, the authors are expected to analyze the relationship between the size of the data and the results of the model be supplemented to support this statement.

Thank you for your valuable feedback regarding the relationship between the size of the dataset and the performance of different models, as mentioned in our Conclusion section. In response to your suggestion, we have decided to enrich our conclusion with pertinent literature that delves into this aspect. We plan to supplement our conclusion with references to existing literature that thoroughly explores this aspect, although we did not perform this analysis ourselves. Reference 52 explores the performance of different classification models for analyzing datasets with different sample sizes. By referencing authoritative studies, we aim to substantiate our claims and offer a more comprehensive understanding of the dynamics between data volume and model efficacy.

  1. The authors are expected to provide methods for the visualization of the specific results of classification using the model.

We appreciate the reviewer's suggestion to provide methods for the visualization of specific classification results derived from our model. Understanding the significance of visual representation in clarifying the model's performance, we have incorporated Figure 4 in Section 2.4. It presents a chart comparing two incorrectly identified samples: (a) S. lugdunensis and (b) E. coli2. This comparison uses a weighted distance metric and a grouped bar chart to show their similarity in our 30-class model.  Conversely, spectra that are correctly identified usually show a negative distance for the specific class they belong to, while distances for all other classes are positive and significantly larger. The confusion matrix of all methods is provided in the supplementary information, table S2. This visualization approach, including the introduction of Figure 4 and the confusion matrix in the supplementary information, is intended to offer a more intuitive and comprehensive understanding of the classification outcomes, thereby enhancing the interpretative clarity of our study's findings.

  1. In the beginning of the Results, a figure of workflow can be provided to visualize the logic of this study.

We acknowledge the reviewer's insightful recommendation to include a workflow diagram at the beginning of the Results section to visually communicate the logical structure of our study. Recognizing the value of a graphical representation in enhancing comprehension and flow, we have accordingly integrated such a visualization, a workflow (Figure1) of the study is added to the “Results and Discussion” Section 1. It is aimed at providing a clear and concise graphical summary of our study's methodology, thereby facilitating a more intuitive understanding of the research process and findings.

  1. In the introduction, the authors are encouraged to include more examples of CNN applied to the field of chemistry and life sciences. One good example is the study completed by the Rajan group (https://doi.org/10.1021/acs.jcim.0c01409)

Thank you for the recommendation to enrich the introduction of our paper with more examples of CNN applications in the field of chemistry and life sciences. The inclusion of such pertinent studies, including the notable work by the Rajan group, indeed strengthens the contextual framework and relevance of our research. In response to your valuable suggestion, we have included the Rajan group study and additional references in the introduction to provide more examples of CNN applications in chemistry and life sciences. We trust that the addition of the Rajan group study, along with other relevant references, will provide a solid foundation for the ensuing sections of our study.

Reviewer 2 Report

Comments and Suggestions for Authors

The authors describe the use of a new Siamese chemometric modeling procedure to identify bacterial species based on their Raman signals.  Though, I believe much work in revision is needed on this manuscript by addressing the items below.

Line 76-77: It will be good to explain how a classification problem and a similarity problem differ.  This is not encountered regularly in the chemo metric literature.  This may be better explained using  a figure to show the architecture of the Siamese network.

Line 81: How is the similarity metric calculated?  Perhaps this could be included with the additional explanation (or summary figure) above.

Line 96: At this point in the manuscript, it is unclear how Siamese-model1 and Siamese-model2 differ.

Table 1:  What about specificity, in addition to sensitivity?  This is an important calculation that indicates false positives of the model.  I believe this is critical to include when assessing model performance.

Table 1: There are 5420 spectra.  However, some of the models contain many times that number of parameters.  How is this justified?  It is very common practice that a model should not contain more parameters than number of data points involved in training/testing.  If this cannot be answered adequately, I must recommend the manuscript be rejected during re-review.

Figure 1: From the figure caption, it is unclear if this is for model training or model testing data.  The text mentions this is testing data (line 179).  This should be added to the caption.  It appears that sensitivity/specificity of predictions can be calculated for each bacterial species.  This may be more effective, as some species may be easier than others to identify.  For those found more difficult to identify, what do they have in common with the mis-identified strain?  For example, E. coli 1 was often mis-identified as E. coli 2.  This is understandable and could lead to a better overall result than 72% overall accuracy.

Figure 1: As this figure is presented in the manuscript, it is unclear what is the difference between strains of the same species (E. coli 1 vs. E. coli 2).  Why not include both under a composite E. coli single class?  Strain differentiation should be much more difficult than species differentiation.

Line 181: The reference to the F1 score is unclear.

After finishing the manuscript, I am still unsure of how E. coli 1 differs from E. coli 2 —— And the many other instances of this strain identification.  The real strain identifiers should be used here.  This is important because I do not know how E. coli 1 and 2 differ (for example, pathogen vs. non-pathogen?).

Lines 237-41: Provides some references here.

Figure 5 and Table 2 need a new explanation in the manuscript.  I have considerable experience in the field, and it is not clear to me what the authors are trying to describe.  It is also unclear how the models were implemented.  Is this from a Python library?  Where are these models available?  How could this work be replicated?

Author Response

We want to express our gratitude for providing us with the chance to enhance the quality of our submitted manuscript. The reviewers constructive and insightful comments were greatly appreciated. Please find attached our detailed responses to all queries, addressing each point raised. We have put forth our best efforts to enhance the manuscript and sincerely value your diligent input. We hope that our revisions will meet with your approval. 

Reviewer2:

The authors describe the use of a new Siamese chemometric modeling procedure to identify bacterial species based on their Raman signals.  Though, I believe much work in revision is needed on this manuscript by addressing the items below.

  1. Line 76-77: It will be good to explain how a classification problem and a similarity problem differ. This is not encountered regularly in the chemometric literature.  This may be better explained using a figure to show the architecture of the Siamese network.

We are grateful for your suggestion to elucidate the distinction between a classification problem and a similarity problem, particularly noting its rarity in chemometric literature. Understanding the importance of clarity in these foundational concepts, we have addressed the request with the addition of text clarifying the difference between a classification problem and a similarity problem. We also changed figure 8 that shows the architecture of the Siamese network to visually illustrate the differences for better understanding. Figure 8 caption also explained about a similarity problem. With the inclusion of a detailed textual explanation and the enhanced visual representation in Figure 8, we aim to provide a clear and comprehensive understanding of the two problem types and their respective implications in our study's context.

  1. Line 81: How is the similarity metric calculated? Perhaps this could be included with the additional explanation (or summary figure) above.

Thank you for your inquiry about the computation of the similarity metric within the Siamese network framework, as mentioned in line 81. The process of calculating this metric is indeed fundamental to the network's functionality and merits a detailed explanation. To address this, we have elaborated on the mechanism of the learnable weighted distance metric. As mentioned in line 277, the network extracts feature vectors f1 and f2 for the two input spectra. The network then calculates the absolute difference between these feature vectors, obtaining ‖f1 - f2‖. Unlike traditional fixed metrics (like Euclidean distance), this network employs a learnable weighted distance metric. This means the network learns a set of weights w during training. These weights are not static but are adjusted based on the data and the task. The network computes the weighted distance as d = w^T ∙ ‖f1 - f2‖. The network is learning the importance (or weight) of each dimension in the feature vectors in terms of contributing to the overall distance measure.

We include Figure 4 in Section 2.4; It presents a chart comparing two incorrectly identified samples: (a) S. lugdunensis and (b) E. coli2. This comparison uses a weighted distance metric and a grouped bar chart to show their similarity in our 30-class model.  Conversely, spectra that are correctly identified usually show a negative distance for the specific class they belong to, while distances for all other classes are positive and significantly larger.

Furthermore, to ensure a comprehensive understanding, we have included Figure 4 in Section 2.4. This figure serves to visually demonstrate the application of the weighted distance metric in distinguishing between correctly and incorrectly identified samples, thereby offering an intuitive grasp of the metric's practical implications. Through this textual and visual elaboration, we aim to provide a clear explanation of how the similarity metric is connected to the functioning of the Siamese network in our study.

  1. Line 96: At this point in the manuscript, it is unclear how Siamese-model1 and Siamese-model2 differ.

We acknowledge the need for clarity regarding the differentiation between Siamese-model1 and Siamese-model2 as indicated in line 96. Understanding the significance of delineating these models distinctly for the readers, we have compared their architectures in Table 2, line 268 and section 3.4. Siamese-model1 is simpler, with a single dense layer leading to a sigmoid activation for binary classification. This simplicity offers robustness and efficiency, making it suitable for scenarios with limited data or computational resources. On the other hand, Siamese-model2 is more complex, with a sequence of three dense layers that increase its capacity to learn more patterns in the data.  

  1. Table 1: What about specificity, in addition to sensitivity?  This is an important calculation that indicates false positives of the model.  I believe this is critical to include when assessing model performance.

We truly appreciate your valuable suggestion to include specificity in addition to sensitivity in Table 1. Recognizing the critical role of specificity in offering a comprehensive understanding of model performance, particularly in terms of its ability to accurately identify negatives (false positives), we have included the specificity in Table 1 for all models for better comparison. 

  1. Table 1: There are 5420 spectra. However, some of the models contain many times that number of parameters.  How is this justified?  It is very common practice that a model should not contain more parameters than number of data points involved in training/testing.  If this cannot be answered adequately, I must recommend the manuscript be rejected during re-review.

We understand and appreciate the concern raised regarding the number of parameters in the models relative to the dataset size, as pointed out in Table 1. The question of overfitting is indeed crucial in the context of model training and generalizability. To address this valid point, we have to differentiate between classical statistical modeling and Convolutional Neural Networks (CNNs). In classical statistical modeling, having more model parameters than data points is a concern, as it can lead to overfitting. However, CNNs do not follow this convention, often having millions of parameters yet effectively generalizing. In a CNN, a high number of parameters can be zero or near zero, especially with L2 regularization. CNNs employ parameter sharing and local connectivity, where weights are reused across input parts (wavenumber). This setup significantly reduces the number of independent parameters.

Deep learning models counteract overfitting through early stopping and regularization techniques such as dropout, weight decay, and batch normalization. These strategies limit the model's capacity to memorize the training data. In the case of 30 bacteria, transfer learning reduces the need for training data by providing a robust starting point. This is achieved by pre-training a model on larger dataset and fine-tuning it for a specific task.

However, when the training set is very small, learning is difficult. Siamese networks can present an advantage that consists of solving a simpler problem, which is a similarity problem and not classification, that is, whether two signals are similar. And not if it belongs to one of the predefined categories, then conceptually, although the architecture is defined, the problem to be optimized is simpler and tends to memorize less and can escape overfitting.

However, training becomes challenging with very limited data. In such scenarios, Siamese networks offer a unique advantage by solving a simpler task: assessing the similarity between two spectra rather than categorizing each spectrum into predefined classes. This approach simplifies the optimization problem. Although the architecture remains complex, the task is more straightforward. Consequently, the network tends to memorize less, reducing the chance of overfitting and enhancing its generalization ability, even with minimal data.

Through this explanation, we aim to justify the architecture of our models and the strategies employed to mitigate overfitting, ensuring that despite the high parameter count, the models are robust, generalize well, and are defined to effectively handle the specific characteristics of our dataset.

  1. Figure 1: From the figure caption, it is unclear if this is for model training or model testing data. The text mentions this is testing data (line 179).  This should be added to the caption.  It appears that sensitivity/specificity of predictions can be calculated for each bacterial species.  This may be more effective, as some species may be easier than others to identify.  For those found more difficult to identify, what do they have in common with the mis-identified strain?  For example, E. coli 1 was often mis-identified as E. coli 2.  This is understandable and could lead to a better overall result than 72% overall accuracy.

Figure 1: As this figure is presented in the manuscript, it is unclear what is the difference between strains of the same species (E. coli 1 vs. E. coli 2).  Why not include both under a composite E. coli single class?  Strain differentiation should be much more difficult than species differentiation.

We appreciate your thorough examination of Figure 1 and the thoughtful insights regarding species differentiation and the clarity of the figure caption. Recognizing the importance of precise differentiation of species classification, we have updated the captions in Figure 1 and Figure 2, which refer to testing data, and where the sensitivity and specificity of predictions can be calculated for each bacterial.

We include in the supplementary material Table S1, which describes the species name, figure label, and isolate code. Differentiating E. coli strains, such as BDMS T4169 and Seattle 1946, is essential in laboratory and clinical settings. While classifying them under a single E. coli category may seem convenient, it is crucial to recognize their unique genetic and phenotypic characteristics. These strains exhibit distinct behaviors, toxin production capabilities, and antimicrobial resistance profiles, essential for quality control and specific testing applications. For instance, BDMS T4169 is crucial for BBL chromagar effectiveness, while Seattle 1946 serves broader quality control purposes.

As an illustration, Figure S3 shows the mean spectra from training data for three bacterial strains, including E. coli1 and E. coli2. We can observe that the patterns may be similar between E. coli1 and E. coli2 in some sections, but there are marked differences. Similarly, E. coli1 and K. aerogenes seem more similar to each other than E. coli2, but several differences are also noticeable.

Finally, the decision to maintain separate classes for E. coli 1 and E. coli 2 rather than combining them into a composite 'E. coli' class aligns with the configuration established by Ho, Chi-Sing, et al. "Rapid identification of pathogenic bacteria using Raman spectroscopy and deep learning." Nature Communications 10.1 (2019): 4927, which introduced the dataset.

By updating the caption to Figure 1 and Figure 2 and elaborating on the rationale behind the differentiation of strains, such as E. coli 1 and E. coli 2, we aim to provide a clear and comprehensive understanding. The inclusion of supplementary material, like Table S1 and Figure S3, further aids in illustrating the subtle yet significant distinctions among bacterial strains.  

  1. Line 181: The reference to the F1 score is unclear.

We acknowledge the reviewer's comment regarding the need for clarity in the reference to the F1 score in line 181. Recognizing the importance of a clear and comprehensive presentation of the performance metrics, we have addressed this concern by including a detailed definition and the equation of the F1 score.  An average sensitivity (recall) of 72.0% and an average precision of 76.1% indicate the model's performance in correctly identifying positive samples and the precision of its positive predictions across all 30 classes. An average F1 score of 71% represents the harmonic mean of precision and recall, providing a balanced view of the model's overall performance. This F1 score suggests that, on average, the model is effective in both correctly identifying positive samples and making precise positive predictions across all 30 classes. This addition is intended to enhance the clarity and completeness of our presentation, thereby offering a more nuanced understanding of the model's efficacy in classification tasks.

  1. After finishing the manuscript, I am still unsure of how E. coli 1 differs from E. coli 2 —— And the many other instances of this strain identification. The real strain identifiers should be used here.  This is important because I do not know how E. coli 1 and 2 differ (for example, pathogen vs. non-pathogen?).

We understand the reviewer's concern for clarity regarding the differentiation between E. coli 1 and E. coli 2, as well as the specificity of strain identification throughout the manuscript. Recognizing the importance of providing clear and precise strain identifiers to elucidate their distinct characteristics, we have included in the supplementary material Table S1, which describes the species name, figure label, and isolate code. The decision to maintain separate classes for E. coli 1 and E. coli 2 rather than combining them into a composite 'E. coli' class aligns with the configuration established by Ho, Chi-Sing, et al (2019), which introduced the dataset.  By including detailed descriptions in the supplementary material Table S1 and referencing the original dataset configuration as per Ho, Chi-Sing, et al. (2019), we aim to clarify the specificities of each bacterial strain.

  1. Lines 237-41: Provides some references here.

We appreciate the reviewer's guidance in enhancing the credibility and depth of our manuscript by suggesting the inclusion of references in lines 237-41. Acknowledging the significance of grounding our statements with scholarly sources, we have responded by incorporating relevant references to support and reinforce the content. This addition is intended to provide readers with additional sources for further exploration and validation of the points discussed in this section.

  1. Figure 5 and Table 2 need a new explanation in the manuscript. I have considerable experience in the field, and it is not clear to me what the authors are trying to describe.  It is also unclear how the models were implemented.  Is this from a Python library?  Where are these models available?  How could this work be replicated?

We appreciate the reviewer's valuable feedback regarding Figure 5 and Table 2, as well as the implementation details of the models. Recognizing the importance of clarity, transparency, and replicability in scientific research, we have revised the manuscript to provide a clearer and more comprehensive description of model architectures in this figure and table. We have ensured that their purpose and implications are well explained. The models presented in Figure 5 and the details provided in Table 2 were implemented using TensorFlow in Python. To enhance transparency and facilitate replication of our work, we are planning to make the models, along with the code and relevant documentation, available in a dedicated GitHub repository. This will provide a detailed guide for users to replicate our experiments and better understand the implementation. Our aim is to make the methodology transparent and accessible to facilitate the replication and further exploration of our work. By revising the manuscript to provide a clearer explanation of model architectures, specifying the use of TensorFlow in Python for implementation, and confirming our intention to make the models and code available in a GitHub repository, we aim to enhance the accessibility and transparency of our work, allowing other researchers to replicate and build upon our experiments with ease.

Reviewer 3 Report

Comments and Suggestions for Authors

Raman spectroscopy combined with neural network for clinically relevant bacterial classification aspects is presented in this manuscript. From the results presented so far, compared to models such as CNN, the two Siamese models constructed do not show any significant improvement in either modelling speed or model prediction predictive performance. The following are some comments on this manuscript.

1. In the introduction, the advantages of Raman spectroscopy over other detection techniques for bacterial analysis should be added. For chemometrics methods, there are many applications for classification, and the authors' presentation here is too one-sided. Moreover, for deep learning methods, it is not only CNN. other methods should be added and their advantages and limitations should be analysed.

2. Experimental samples, a total of 5420 spectra from 6 classes of samples. How many bacteria are in each class and how many spectra were collected from each class? This description needs to be further improved.

3. Model performance comparison, it is suggested to supplement other classical methods, such as PLS-DA, SVM, RF, ANN and so on. In addition, the model performance evaluation index is too single, other classification performance evaluation indexes need to be supplemented.

4. Part 2.2 claims that there are 30 strain sample datasets, but why only 15 class samples appear in the confusion matrix of Figure 1?

5. The Raman spectra of the six bacterial species presented in Figure 3 seem intuitively different, and it is suggested to add some specific illustrations to analyse the differences in the spectra of different samples from a molecular perspective. The same problem applies to Figure 4.

6. It is suggested that the third part of the manuscript be rearranged before the discussion section, which would help the reader.

Comments on the Quality of English Language

Some of the descriptions of the experimental samples need further clarification.

Author Response

We want to express our gratitude for providing us with the chance to enhance the quality of our submitted manuscript. The reviewers constructive and insightful comments were greatly appreciated. Please find attached our detailed responses to all queries, addressing each point raised. We have put forth our best efforts to enhance the manuscript and sincerely value your diligent input. We hope that our revisions will meet with your approval. 

Reviewer3:

Raman spectroscopy combined with neural network for clinically relevant bacterial classification aspects is presented in this manuscript. From the results presented so far, compared to models such as CNN, the two Siamese models constructed do not show any significant improvement in either modelling speed or model prediction predictive performance. The following are some comments on this manuscript.

  1. In the introduction, the advantages of Raman spectroscopy over other detection techniques for bacterial analysis should be added. For chemometrics methods, there are many applications for classification, and the authors' presentation here is too one-sided. Moreover, for deep learning methods, it is not only CNN. other methods should be added and their advantages and limitations should be analysed.

We are grateful for your insightful comments and recommendations regarding the introduction section of our manuscript. Recognizing the importance of providing a balanced and comprehensive overview, we have made improvements in the introduction by including a section on the advantages of Raman spectroscopy over other detection techniques for bacterial analysis. Additionally, we acknowledge the need for a more balanced presentation of chemometrics methods, addressing various applications for classification. Therefore, more machine learning and deep learning methods and their respective advantages and limitations were introduced. We appreciate your input and believe these additions strengthen the manuscript.

These enhancements to the introduction aim to provide a better understanding of the advantages of Raman spectroscopy, the versatility of chemometrics methods, and the diversity of machine learning and deep learning approaches, along with their respective advantages and limitations.  

  1. Experimental samples, a total of 5420 spectra from 6 classes of samples. How many bacteria are in each class and how many spectra were collected from each class? This description needs to be further improved.

Thank you for your feedback regarding the description of the experimental samples. We have taken steps to improve the clarity and detail provided in the manuscript. Specifically, we have addressed your concern by including the number of spectra per class in section 2.1, as well as providing more comprehensive information about the dataset in section 3.1. By incorporating these additional details, we aim to provide a clearer and more informative description of the dataset.

  1. Model performance comparison, it is suggested to supplement other classical methods, such as PLS-DA, SVM, RF, ANN and so on. In addition, the model performance evaluation index is too single, other classification performance evaluation indexes need to be supplemented.

We appreciate your suggestion to supplement the comparison with other classical methods and include additional classification performance evaluation indexes. To address these valuable points, we have included the results of PLS-DA, PCA-SVM and PCA-RF to the text and Table 1. The classification parameters, sensitivity and specificity are also calculated and added to Table 1. With these changes, we aim to provide a more comprehensive and balanced evaluation of model performance, allowing readers to better assess the comparative effectiveness of the methods employed in our study.

  1. Part 2.2 claims that there are 30 strain sample datasets, but why only 15 class samples appear in the confusion matrix of Figure 1?

We appreciate your keen observation and valuable input regarding the discrepancy between the mention of 30 strain sample datasets and the appearance of only 15 class samples in the confusion matrix of Figure 1. To address this inconsistency and provide a clearer explanation of our methodology, we have revised the manuscript for greater clarity. Now, in section 2.2, we detail how we utilized half of the classes from a 30-class Raman spectra dataset to improve the Siamese network's embedding visualizations, as displayed in the Supplementary material. Further, in section 2.3, we describe our evaluation of the methodology's performance on all 30 classes, considering both balanced and imbalanced data scenarios. Through these revisions, we aim to enhance the clarity and coherence of our manuscript, ensuring that readers have a more accurate understanding of our dataset utilization and evaluation approach in both balanced and imbalanced scenarios.

  1. The Raman spectra of the six bacterial species presented in Figure 3 seem intuitively different, and it is suggested to add some specific illustrations to analyze the differences in the spectra of different samples from a molecular perspective. The same problem applies to Figure 4.

Thank you for your suggestion to provide specific illustrations to analyze the differences in Raman spectra from a molecular perspective. We have taken your feedback into consideration. In Figure 3 and Figure 4 the mean spectra may seem intuitively different for each of the bacterial species, however, the intra class variance is high. As illustrated in Figure S3, we present the mean spectra from training data for three bacterial strains, including E. coli1 and E. coli2. It's evident that there are similarities in some sections between E. coli1 and E. coli2, but there are also notable differences. Similarly, E. coli1 and K. aerogenes appear more similar to each other than to E. coli2, although several differences are still noticeable. The decision to maintain separate classes for E. coli 1 and E. coli 2, rather than combining them into a single 'E. coli' class, aligns with the configuration established by Ho, Chi-Sing, et al. in their paper titled "Rapid identification of pathogenic bacteria using Raman spectroscopy and deep learning" published in Nature Communications (2019). This configuration was introduced in the dataset and serves as a reference for our approach. We include in the supplementary material Table S1, which describes the species name, figure label, and isolate code. Through these additions, we aim to provide readers with a more detailed and molecular-level perspective on the differences in Raman spectra between bacterial strains, particularly E. coli1 and E. coli2. The inclusion of Figure S3 and reference to Table S1 in the supplementary material will allow readers to delve deeper into the spectral distinctions and gain a better understanding of the rationale behind maintaining separate classes for E. coli 1 and E. coli 2.

  1. It is suggested that the third part of the manuscript be rearranged before the discussion section, which would help the reader.

We appreciate your suggestion to rearrange the third part of the manuscript before the discussion section for improved readability. However, we regret to inform you that the current order of sections aligns with the format requirements of the journal in which our manuscript is being submitted. As such, we are unable to make changes to the order of sections that deviate from the journal's guidelines. We apologize for any inconvenience this may cause and hope that the organization of the manuscript remains coherent and understandable for readers within the journal's format constraints.

Round 2

Reviewer 3 Report

Comments and Suggestions for Authors

This manuscript can be accepted for publication in the current version.